# Imaginary Speech Recognition Using a Convolutional Network with Long-Short Memory

Ana-Luiza Rusnac * and Ovidiu Grigore *

Department of Applied Electronics and Information Engineering, Faculty of Electronics, Telecommunications and Information Technology, Polytechnic University of Bucharest, 060042 Bucharest, Romania

* Correspondence: ana_luiza.dumitrescu@upb.ro (A.-L.R.); ovidiu.grigore@upb.ro (O.G.)

**Abstract:** In recent years, a lot of researchers' attentions were concentrating on imaginary speech understanding, decoding, and even recognition. Speech is a complex mechanism, which involves multiple brain areas in the process of production, planning, and precise control of a large number of muscles and articulation involved in the actual utterance. This paper proposes an intelligent imaginary speech recognition system of eleven different utterances, seven phonemes, and four words from the Kara One database. We showed, during our research, that the feature space of the cross-covariance in frequency domain offers a better perspective of the imaginary speech by computing LDA for 2D representation of the feature space, in comparison to cross-covariance in the time domain and the raw signals without any processing. In the classification stage, we used a CNNLSTM neural network and obtained a performance of 43% accuracy for all eleven different utterances. The developed system was meant to be a subject's shared system. We also showed that, using the channels corresponding to the anatomical structures of the brain involved in speech production, i.e., Broca area, primary motor cortex, and secondary motor cortex, 93% of information is preserved, obtaining 40% accuracy by using 29 electrodes out of the initial 62.

**Keywords:** imaginary speech recognition; electroencephalography; long-short term memory neural network; convolutional neural network; cross-covariance features; speech brain areas

## 1. Introduction

Imaginary speech recognition (ISR) systems have grown in popularity in recent years. These types of systems usually aim to acquire signals from the brain using invasive or non-invasive methods, preprocess them to improve the quality of the signal, extracts the features with the main purpose of concentrating the significant information into smaller data, and finally, classifies the features in order to offer feedback to the user. This processing chain that leads to obtaining the final system must work in real time on portable devices to be actually used in everyday activities by the people who need it. The main purpose of ISR is to come to the aid of persons who, after suffering some disorders (afflictions,) such as the lock-down syndrome, cerebral palsy, etc., lost the ability to speak.

An important part of the ISR systems is represented by the signals acquisition stage. There are several methods of collecting the signals for a brain computer interface (BCI) system, such as electroencephalography (EEG), electrocorticography (ECoG), magnetoencephalography (MEG), positron emission tomography (PET), and functional magnetic resonance imaging (fMRI). The most common method used is EEG. The main advantage of the EEG signals is the fact that the acquisition is non-invasive and low-cost, compared to MEG, PET, and fMRI. However, since the signals are acquired from the surface of the scalp, there are multiple layers between the source and the electrode, which leads to a high attenuation of the signal, and it is more sensitive to noises. The second most used acquisition method for imaginary speech decoding is ECoG, due to the higher quality of the collected data, since the signals are acquired directly from the cortex (invasive).

Nevertheless, this advantage also brings the disadvantage of being an invasive method, leading to difficulties in data acquisition.

## 2. State of the Art

In recent years, recognizing silent speech from cortical signals attracted more and more attention of the researchers. Different approaches were tried over the years, aiming to achieve the best performances.

The first attempts consisted of creating the desired words from letters using different methods of choosing the letters, such as moving a cursor on a monitor [1] or following a matrix of ASCII characters as the lines and columns are highlighted [2]. A successful real-time communication system, based on creating the words letter-by-letter, was developed by Ujwal Chaudhary et al. [3] on a patient with amyotrophic lateral sclerosis (ASL). Due to the degradation of the motor functions, the only way of communication for the patient was by using brain signals. The researchers implanted a 64-microelectrode array, which allowed the subject to modulate neural firing rates after receiving audio feedback with a block of letters first and each letter from the block after to create words. After days of training, on the 245th day after implantation, the patient was able to create complex sentences, such as, "I would like to listen to the album by Tool loud", with a mean of one character per minute.

These methods offered significant results and have the advantage of improving over time as the subjects practice mastering the system. However, this type of communication is difficult, unnatural, and takes relatively a lot of time to form a word (one character per minute [3]).

Other approaches targeted decoding the speech directly from the imagined words or phonemes. These methods assume that, during imaginary speech, the brain has different marks of electric activity, according to different pronunciations of each word.

Trying to find different marks of imaginary speech, Thimotee Proix et al. [4] investigated the ECoG signals acquired during overt and imagined speech. In the conducted study, the researchers first tried to find similarities between overt and imagined speech. They computed the power spectrum of four frequency bands, theta (4–8 Hz), low-beta (12–18 Hz), low-gamma (25–35 Hz), and BHA (80–150 HZ), and they observed that the BHA power for both overt and imagined speech increased in the sensory and motor regions, while beta power decreased over the same regions. Nevertheless, when trying to differentiate different syllables in binary classes, articulatory, phonetic, and vocalic, the results for imaginary speech significantly dropped, compared to overt speech in the BHA band. Better performance was recorded when using power in the beta band, but with room for improvement. Significant improvements of the results were obtained when the trials were aligned on recorded speech onset, which was impossible to do for the imaginary speech.

Other study based on ECoG signals managed to obtain important results when classifying five different words in a **patient-specific system** [5]. The researchers used high-gamma time features, which were aligned using dynamic time warping (DTW), and finally obtained a matrix feature by computing the DTW-distance between the realigned trials. These features, combined with support vector machine (SVM), offered encouraging results for a five-class classification task, obtaining a 58% mean accuracy for all five subjects. In 2019, Miguel Angrick et al. [6] managed to synthesize speech from **ECoG signals** by also using the **speech signal** acquired, together with the ECoG signal. The obtained results registered a correlation between the reconstructed speech signal and the actual speech, higher than the chance for all the analyzed subjects. Moreover, one of the subjects had a correlation of 0.69, which allowed for a good reconstruction of the vocal signal.

Recent studies of stereoelectroencephalography (sEEG) signals regarding decoding the imaginary speech reported the development of a real-time system with speech synthesis for only one patient, due to the difficulty of data acquisition [7]. The researchers processed the neural signals by extracting the multichannel high-gamma band as features that were decoded by assigning the neural activity to a mel-scaled audio spectrogram. Further, a regularized LDA-classifier was trained to predict the energy level for each spectral bin.

The results reported an average $0.62 \pm 0.15$ Pearson correlation between the reconstructed signal and the real speech signal, but the reconstructed speech was not yet intelligible.

However, even if ECoG and sEEG offered a good representation of the neural activity with a high signal-to-noise ratio, they were still **invasive** methods of acquiring signals from the brain, which leads to great limitations regarding collecting the signals and being easily accepted by the patients. The alternative to these invasive methods can be functional magnetic resonance imaging (fMRI), functional near-infrared spectroscopy (fNIRS), magnetoencephalography (MEG), and electroencephalography (EEG). All these methods have their limitations: fMRI is expensive, non-portable, and has a poor temporal resolution, fNIRS takes time to collect the information from the brain (aprox. 2 to 5 s), which makes it harder to aim a real-time system, and MEG is expensive and non-portable. Finally, EEG remains the best option for a real-time noninvasive wearable device, while having the limitation of small amplitude data with a higher signal-to-noise ratio, comparing to ECoG.

In recent years, several studies have used EEG signals for imaginary speech recognition. In study [8], the researchers analyzed six different words: "could", "yard", "give", "him", "there", and "toe", acquired from 15 subjects. In the feature extraction stage, the signals were decomposed using the Daubechies-4 (db4) mother wavelet to eight levels corresponding to the delta, theta, alpha, beta, gamma bands, and additionally, three other bands, <2 Hz, 64–128 Hz and 128–256 Hz, where features such as root mean square, standard deviation, and relative wavelet energy were computed. The features were fed into a random forest (RF) classifier and support vector machine (SVM). The registered results rose above the chance for both classifiers, having a mean accuracy for all 15 subjects of 25.26% RF and 28.61% SVM.

A great breakthrough in the field came along with the open access datasets, such as the Kara One Dataset [9] and Nguyen et al. [10] research dataset. This helped the community to develop research in the imaginary speech field, without having to go through the complicated process of data acquisition, and made it possible to compare the results with other studies developed based on the same environment.

Using the Kara One Dataset, Panachakel et al. [11] managed to develop a **patient-specific** system based on computing a set of statistical features, root mean square, variance, kurtosis, skewness, and the 3rd order moment, over the obtained signals after decomposition using db4 mother wavelet into seven levels. Afterwards, the signals were fed into a deep learning architecture with two layers of 40 neurons. The results offered a 57.15% mean accuracy for all subjects, significantly higher compared to the results obtained without using deep learning. With a more complex deep learning architecture, the researchers from University of British Columbia managed, in their paper [12], to present a generalized system for imaginary speech **binary classification,** achieving the best accuracy of 85.23% for discrimination between consonant and vocal phonemes. The results were achieved using the covariance matrix between the channels as a feature matrix and a combination between convolutional neural networks (CNN), long-short term memory (LSTM) neural networks, and a deep autoencoder (DAE).

In 2021, a larger dataset was collected in Moscow, Russia, containing signals acquired from 270 healthy subjects during the silent speech of eight Russian words: "forward", "backward", "up", "down", "help", "take", "stop", and "release". The **patient-specific** developed system offered an accuracy of 84.5% for all nine words classification and 87.9% for **binary classification** using the deep learning architecture of ResNet18, in combination with two layers of gated recurrent units (GRUs). Finally, the researchers claimed that there are strong differences between the signals from different subjects, and it is more plausible to develop a patient-specific system with a high accuracy than a generalized one.

Deep learning architectures showed a better performance of imaginary speech signals and gained popularity over the years. Recently, EEG studies started to concentrate on the LSTM neural network, due to the advantages it has for a continuous series. The LSTM neural network offered significant results on applications such as epilepsy prediction [13] and imaginary speech recognition [14]. The results obtained by the researchers in [14]

registered a maximum accuracy of 73.56% for a subject-independent system and a **four-class** problem: "sos", "stop", "medicine", and "washroom".

In this paper, we focused on developing a subject's shared system for imaginary speech recognition. By the subject's shared system, we mean that one system was developed for all registered users in the database. However, this system was different from a patient-specific system in such a matter that, when introducing a new subject in the database, it will only need a fine-tuning of the neural network and not an entire subject training. During our research, we used Kara One database, which was preprocessed for further usage. Cross-covariance in frequency domain was computed in the feature extraction stage, and all the signals were further introduced into a CNNLSTM neural network. We showed that CNNLSTM performed better than the CNN neural network by increasing the accuracy from 37% to 43%. In the paper, the system behavior after reducing the number of channels and selecting the channels from the main brain areas and the areas corresponding to the anatomical structures involved in imaginary speech production was also tested. We showed that the channels corresponded to the anatomical structures involved in speech production concentrates 93% of information, obtaining a maximum accuracy of approx. 40%. Even if the accuracy dropped by 3%, a lot more was gained in terms of time of execution, comfort, portability, and costs.

The system was developed using python as programming language, and the CNNLSTM neural network was developed using TensorFlow, an end-to-end open-source platform for machine learning [15].

## 3. Materials and Methods

### 3.1. Preparing Database

The Kara One Database was developed following a partnership between the University of Toronto and Toronto Rehabilitation Institute [9]. It contains signals acquired from eight male and four female participants (a total of twelve subjects), all right-handed, with no visual, hearing, or motor problems. All signals were collected using a standard protocol consisting of a (1) rest state, (2) stimulus state, in which the prompt with the stimulus appeared, (3) a five-second of stimulus imagined speaking state, and (4) an actual articulation of the stimulus. An important aspect of this database consisted of randomly presenting the stimulus to the subject at each epoch, avoiding the accidental creation of temporal correlations in EEG signals, as presented in [16]. The prompts contained seven phonemes: /iy/, /uw/, /piy/, /tiy/, /diy/, /m/, /n/ and four words chosen from the Kent's list of phonetically similar pairs: "pat", "pot", "knew", and "gnaw". The stimulus was chosen to cover a wide range of pronunciation mechanisms, such as nasal: /m/, /n/, "knew", and "gnaw", bilabial: /piy/, "pat", "pot", and /m/, closed vocal tract: /piy/, /tiy/, /diy/, vocals: /iy/, /uw/, and consonants: /m/, /n/. Further, in the study, we only used the signals corresponding to the imaginary speech.

All signals from the database were visually analyzed by a specialist. This process led to elimination of the epochs containing high noises. Additionally, during the visual analysis of the signal, four of the twelve subjects were eliminated from the study, due to the very high noise signals or the presence of unattached ground wires. The number of signals that passed on in the study were of 993 for all subjects and all prompts.

### 3.2. Preprocessing

In the preprocessing stage, the signals were filtered using a notch filter to remove the 60 Hz power line artifact and all multiples of 60 Hz smaller than the Nyquist frequency.

Afterwards, from the 5 s epoch of imaginary speech, corresponding to each utterance, we removed the first and the last 0.5 s corresponding to the transition from one state to another. Further, we segmented the 4 s remaining signal into windows of 0.25 s without overlapping. The window dimension was chosen following a previous study focused, among others, on the analysis of different windows length: 0.25 s, 0.5 s, and 1 s [17]. Finally,

half of the windows were randomly assigned to the training set and the other half to the testing set for the subject's shared system.

### 3.3. Feature Extraction

The feature extraction stage aims to decode the information hidden in the signal and pass the result to a neural network capable of classifying the words and phonemes. In this paper, we used as features the inter-channel covariation in frequency domain. Let $FX^{ch}$ be the fast Fourier transform (FFT) corresponding to the channel $ch$ of the input signal $X^{ch}$, the FFT can be described by the following equation:

$$FX^{ch}(f) = \sum_{t=0}^{n-1} X_t^{ch} e^{-\frac{j2\pi ft}{n}} \tag{1}$$

The cross-covariation between the channels in frequency domain is given by:

$$Cov\left(X^{c1}(t), X^{c2}(t)\right) = E\left[\left[X^{c1}(t) - E(X^{c1}(t))\right]\left[X^{c2}(t) - E(X^{c2}(t))\right]\right] \tag{2}$$

where $X^{c1}(t)$ represents the EEG signal acquired for channel $c1$, $X^{c2}(t)$ is the EEG signal acquired for channel $c2$, and $E[X^{ch}(t)]$ represents the mean value (where $ch$ is corresponding to the specific channel $c1$ or $c2$) and is computed as:

$$E(X^{ch}(t)) = \frac{1}{W} \sum_{i=0}^{W-1} x_i^{ch} \tag{3}$$

The $W$ value of Equation (3) corresponds to the window dimension for which the features are computed. After extracting the features, the resulting matrix for a signal will be of size $[N_{ch} \times N_{ch}]$, where $N_{ch}$ is the number of channels.

The 0.25 s segmented channels were further cropped into 0.1 s window with 50% overlap to preserve the time evolution of the signal, which was passed to the CNN-LSTM architecture. An example of the computed features over 0.1 s window length for each phoneme and word is presented in Figure 1.

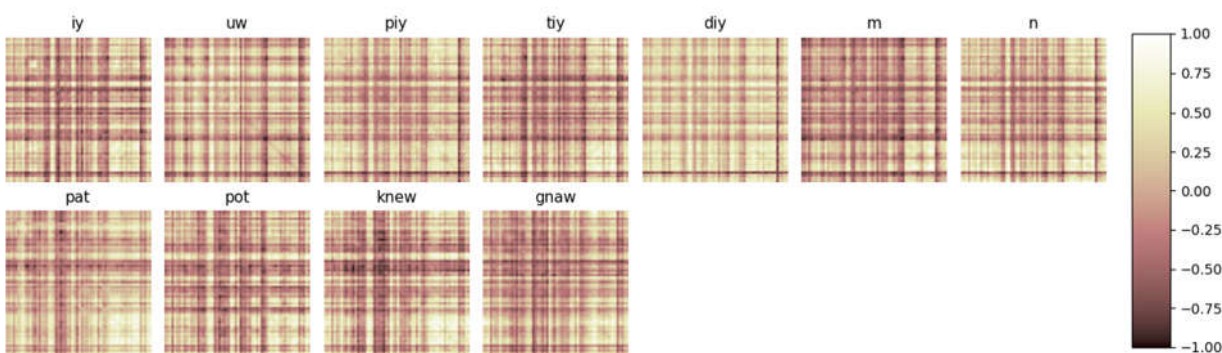

**Figure 1.** Cross-covariance in frequency domain for 0.1 s window for all phonemes and words from the KODB.

Further, we investigated the features by computing the linear discriminant analysis (LDA) to reduce the features' dimensionality to only two components for visual inspection of the computed features. Figure 2 presents a comparison between the results obtained by applying LDA to the computed features cross-covariance in frequency domain (a), the raw signal in time (b), and computed features cross-covariance in time domain (c). It can be easily seen that, when using the frequency domain cross-covariance, the features are very well-structured into separate clusters, compared to the results obtained using the time domain signal or the time domain cross-covariance as features, cases where the classes

overlap and the vectors are mixed in the same region of feature space. Another observation regarding the LDA results is the separability of the phonemes versus words. It can be seen that words and phonemes occupy different regions of the feature space.

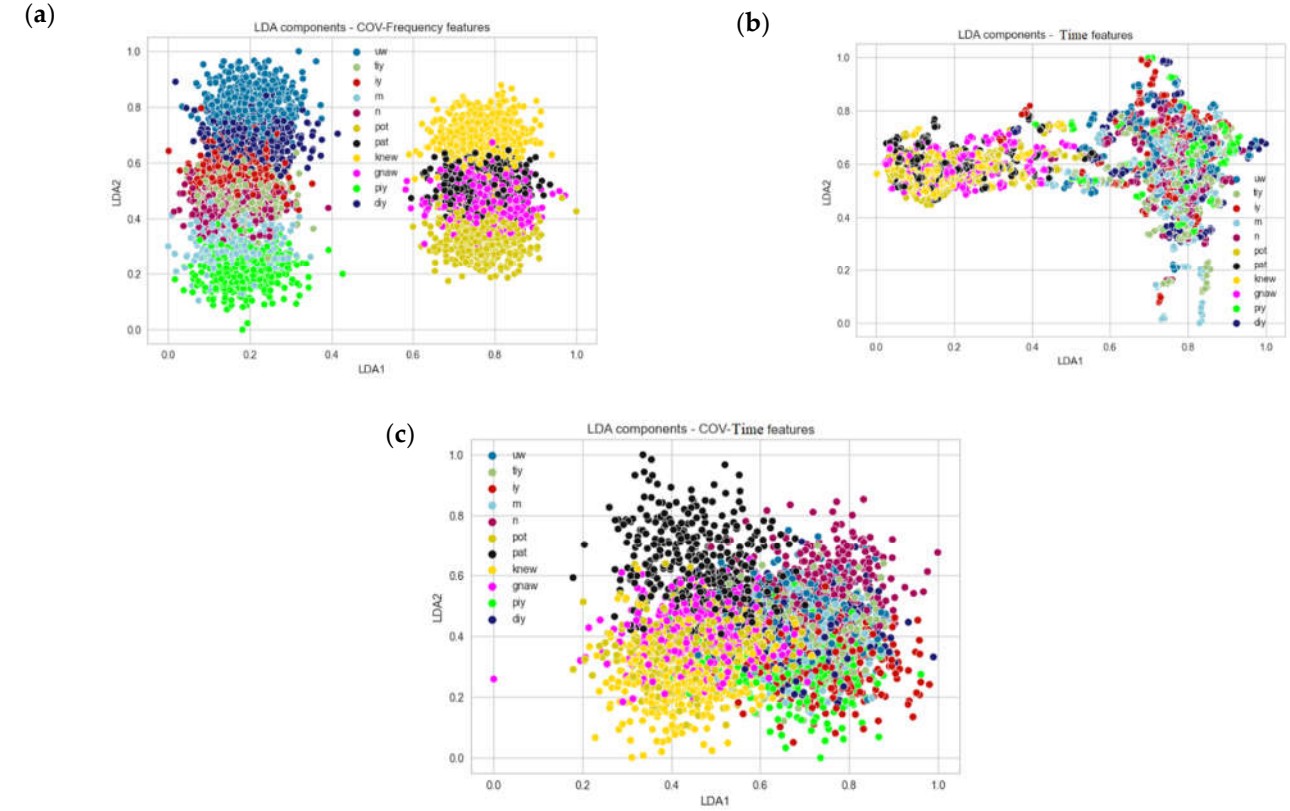

**Figure 2.** Feature reduction using LDA algorithm with eigenvalue decomposition and shrinkage value 0.2 for (**a**) cross-covariance features in frequency domain, (**b**) raw signal in time domain, and (**c**) cross-covariance features in time domain.

In Table 1, the mean and standard deviation of the LDA components for each class were computed to quantify the results observed after the visual analyses. The computed means and the confidence ellipse for each class are pointed out in Figure 3 for all analyses features: cross-covariance in frequency domain (a), raw-signal features (b), and covariance in time domain (c). In time domain, the means were very close to each other, and the standard deviation was high for all classes, each one incorporating a large part of the other classes for phonemes and for words. However, there was a clear differentiation between the words and the phonemes in the feature space. When it came to the cross-covariance in time domain, the distances between the means were getting higher, but the separability between the phonemes and words was lost, too. In the case of cross-covariance in frequency domain, the separability between the phonemes and words was being preserved, and the distances between the means were bigger. However, the classes were still not fully separable.

**Table 1.** The mean and standard deviation of the LDA components for the analyzed features: raw time features, covariance in time domain features, and covariance in frequency domain features.

| | LDA–Time Features | | | | LDA–COV-Time Features | | | | LDA–COV-Frequency Features | | | |
|---|---|---|---|---|---|---|---|---|---|---|---|---|
| | $\mu_{LDA1}$ | $\mu_{LDA2}$ | $\sigma_{LDA1}$ | $\sigma_{LDA2}$ | $\mu_{LDA1}$ | $\mu_{LDA2}$ | $\sigma_{LDA1}$ | $\sigma_{LDA2}$ | $\mu_{LDA1}$ | $\mu_{LDA2}$ | $\sigma_{LDA1}$ | $\sigma_{LDA2}$ |
| /iy/ | 0.7408 | 0.6291 | 0.0969 | 0.1362 | 0.7157 | 0.3545 | 0.0875 | 0.0871 | 0.1773 | 0.5634 | 0.0514 | 0.0584 |
| /uw/ | 0.7656 | 0.6228 | 0.0903 | 0.1126 | 0.6750 | 0.4414 | 0.0748 | 0.0825 | 0.2036 | 0.8149 | 0.0564 | 0.0606 |

**Table 1.** *Cont.*

| | LDA–Time Features | | | | LDA–COV-Time Features | | | | LDA–COV-Frequency Features | | | |
|---|---|---|---|---|---|---|---|---|---|---|---|---|
| | $\mu_{LDA1}$ | $\mu_{LDA2}$ | $\sigma_{LDA1}$ | $\sigma_{LDA2}$ | $\mu_{LDA1}$ | $\mu_{LDA2}$ | $\sigma_{LDA1}$ | $\sigma_{LDA2}$ | $\mu_{LDA1}$ | $\mu_{LDA2}$ | $\sigma_{LDA1}$ | $\sigma_{LDA2}$ |
| /piy/ | 0.7748 | 0.6084 | 0.0903 | 0.1179 | 0.6705 | 0.3351 | 0.0786 | 0.0896 | 0.1891 | 0.2219 | 0.0572 | 0.0604 |
| /tiy/ | 0.7723 | 0.5475 | 0.0737 | 0.1325 | 0.6963 | 0.4130 | 0.0780 | 0.0777 | 0.1739 | 0.4901 | 0.0530 | 0.0586 |
| /diy/ | 0.7578 | 0.6089 | 0.1069 | 0.1283 | 0.6891 | 0.4193 | 0.0805 | 0.0774 | 0.2072 | 0.6897 | 0.0566 | 0.0611 |
| /m/ | 0.7707 | 0.5109 | 0.0867 | 0.1482 | 0.6780 | 0.3877 | 0.0726 | 0.0730 | 0.1686 | 0.3135 | 0.0539 | 0.0655 |
| /n/ | 0.7713 | 0.5744 | 0.0797 | 0.1199 | 0.7112 | 0.5187 | 0.0876 | 0.1016 | 0.1641 | 0.4560 | 0.0542 | 0.0614 |
| pat | 0.1893 | 0.5964 | 0.1035 | 0.0542 | 0.4896 | 0.5855 | 0.0972 | 0.1326 | 0.7696 | 0.5193 | 0.0571 | 0.0569 |
| pot | 0.2141 | 0.5834 | 0.1108 | 0.0545 | 0.4882 | 0.3716 | 0.0990 | 0.0801 | 0.7755 | 0.3395 | 0.0564 | 0.0606 |
| knew | 0.1829 | 0.5765 | 0.0889 | 0.0439 | 0.4761 | 0.3042 | 0.1003 | 0.1005 | 0.7641 | 0.6995 | 0.0574 | 0.0599 |
| gnaw | 0.2092 | 0.5855 | 0.1037 | 0.0481 | 0.5470 | 0.4073 | 0.0978 | 0.0754 | 0.7818 | 0.4635 | 0.0581 | 0.0599 |

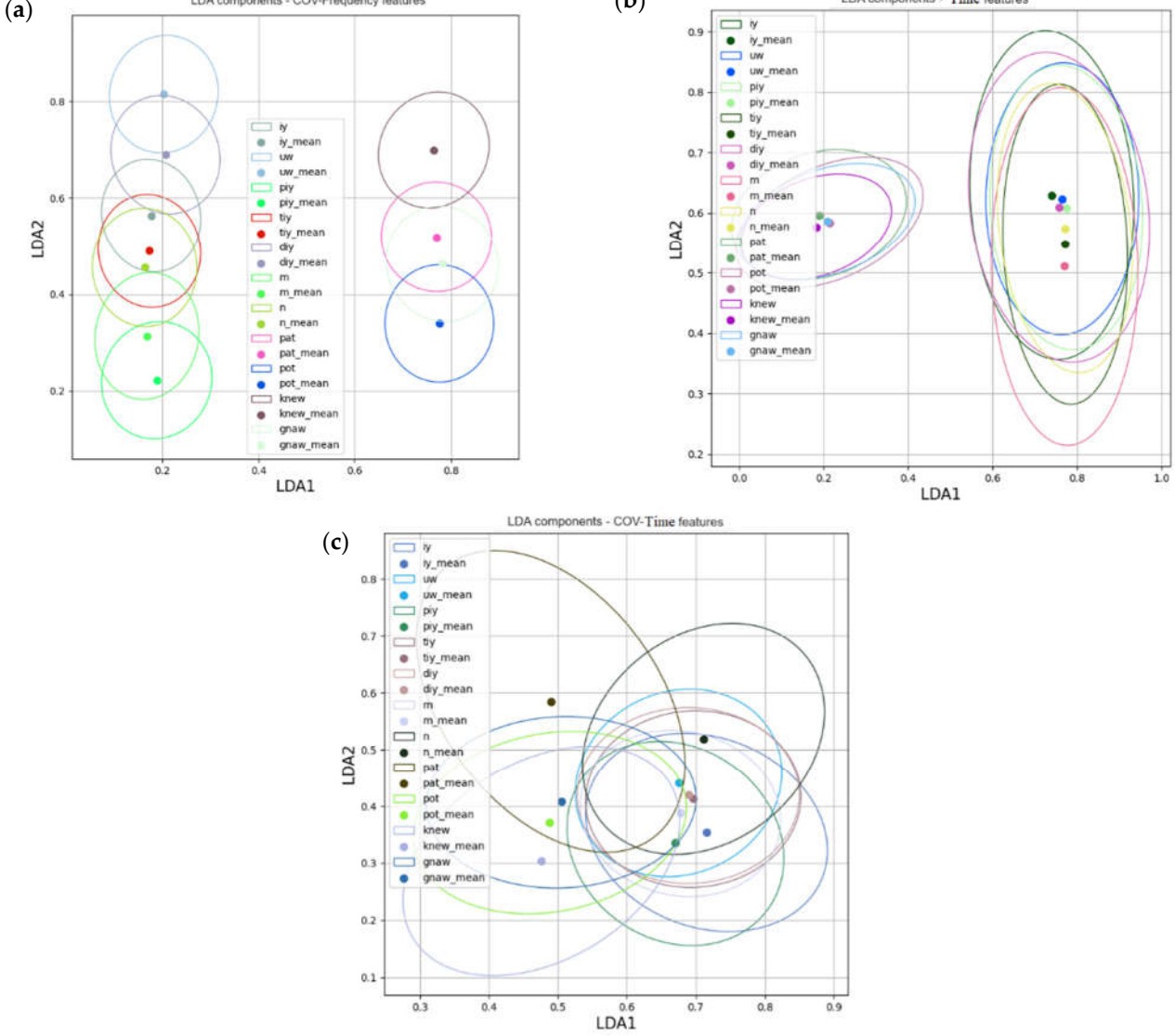

**Figure 3.** The mean of each LDA feature space of KODB classes with the confidence ellipse for feature space analysis of: (**a**) cross-covariance in frequency domain; (**b**) raw-time features; (**c**) cross-covariance in time domain.

After the visual and quantitative analysis of the LDA feature space, we concluded that the cross-covariance in frequency domain offer the best understanding of the imaginary speech signals and was further used in the classification stage.

### 3.4. Classification

LSTM neural networks started to gain more and more popularity when it comes to developing intelligent systems meant to work with time series, such as EEG signals. A wide range of EEG applications exploit the advantages of LSTM consisting of learning long-term dependencies in time series and making connections between different windows of signals. Among these applications, we can list a seizure prediction systems [13], error detections in a musician's performances [18], prediction of customers decisions for brand extension scenarios [19], classification of hand movements [20], and imaginary speech recognition systems [14].

In this paper, we used a 2D convolutional long-short term memory neural network (CNNLSTM) in the classification stage. This neural network added to the advantages of LSTM, i.e., the spatial correlation information between the channels. In other words, this neural network is connecting both the spatial and temporal information of the EEG signals conducting to a better decoding of the brain activity. The detailed information regarding the CNNLSTM layers of the neural networks are explained in [21].

The architecture of the neural network is based on the previous study of the CNN architectures [17], in which the best results were obtained using two convolutional layers with 64 and 128 filters, respectively, connected to a dense layer with 64 neurons. The final layer corresponds to the decision and consists of a dense layer with 11 neurons, the equivalent number of the seven phonemes and four words. During this study, more complex architectures were also tested, but no significant improvements in accuracy were registered.

The size of the input layer corresponds to the size of the features, a matrix of $N \times N_{ch} \times N_{ch} \times N_w$, where $N$ is the number of observations of the training set, $N_{ch}$ is the number of channels used to compute the cross-covariance matrix, and $N_w$ is the number of windows for the LSTM time analysis. The diagram block of the CNNLSTM architecture used in this paper is presented in Figure 4.

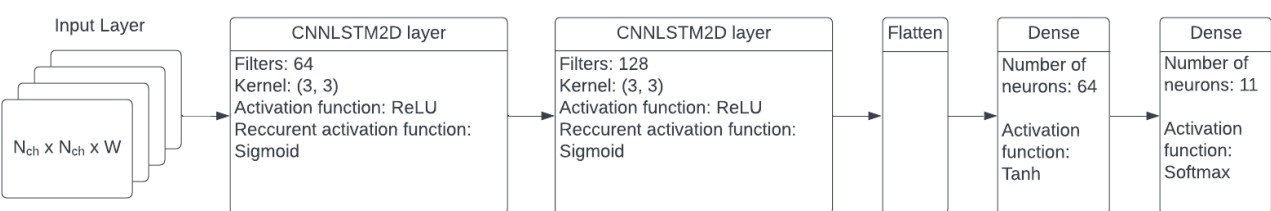

**Figure 4.** Diagram block for the CNNLSTM neural network used in the classification stage.

Finally, the neural network was trained using the optimizer Adam with a learning rate of 0.0001, having as loss function the categorical cross-entropy.

The neural network was developed using TensorFlow with the implemented CNNLSTM2D layer using the algorithm described in [21].

### 3.5. Performance Metrics

The performance metrics used to describe the system's capability of recognizing imaginary speech are implemented in the open-source scikit-learn toolkit detailed in [22].

Let CM be the confusion matrix of the system, described in Table 2, as:

**Table 2.** The confusion matrix of a multiclass dataset; $y_i$ —the desired output of the system, $\hat{y}_i$ —the output of the system. $Nc$—number of classes.

| | | Predicted Label | | | | |
|---|---|---|---|---|---|---|
| | Classes | $c_1$ | $c_2$ | ... | $c_{Nc}$ | Total |
| True label | $c_1$ $y_i \in c_1$ | $\hat{y}_i \in c_1$ | $\hat{y}_i \in c_2$ | ... | $\hat{y}_i \in c_{Nc}$ | $\sum_j^{Nc} C_{c1j}$ |
| | $c_2$ $y_i \in c_2$ | $\hat{y}_i \in c_1$ | $\hat{y}_i \in c_1$ | ... | $\hat{y}_i \in c_{Nc}$ | $\sum_j^{Nc} C_{c2j}$ |
| | $\vdots$ | $\vdots$ | $\vdots$ | $\ddots$ | $\vdots$ | $\vdots$ |
| | $c_{Nc}$ $y_i \in c_{Nc}$ | $\hat{y}_i \in c_1$ | $\hat{y}_i \in c_1$ | ... | $\hat{y}_i \in c_{Nc}$ | $\sum_j^{Nc} C_{Ncj}$ |
| | Total | $\sum_j^{Nc} C_{jc1}$ | $\sum_j^{Nc} C_{jc2}$ | ... | $\sum_j^{Nc} C_{jNc}$ | $\sum_k^{Nc} C_{kk}$ |

The accuracy is computed as the overall performance of the system:

$$Accuracy = \frac{1}{N} \sum_{k=1}^{Nc} C_{kk} \tag{4}$$

where $N$ is the number of input vectors, $Nc$ is the number of classes, and $C_{kk}$ corresponds to the diagonal of the confusion matrix.

The balanced accuracy is used to determine the performance of the system when the imbalanced data is used:

$$Balanced\ accuracy = \frac{1}{Nc} \sum_{k=1}^{Nc} \frac{C_{kk}}{t_k} \tag{5}$$

where $t_k$ is the number of times that class $k$ occurs in the input dataset and is computed as:

$$t_k = \sum_{j=1}^{Nc} C_{jk} \tag{6}$$

*Kappa* coefficient (Cohen's *Kappa*) is generally used to compute the degree of agreement between independent observations and was computed as follows:

$$Kappa = \frac{C\ x\ N - \sum_k^{Nc} p_k \times t_k}{N^2 - \sum_k^{Nc} p_k \times t_k} \tag{7}$$

where $C$ is the total number of correctly predicted elements, and $p_k$ is the number of times that class $k$ was predicted, and they are described by the following equations:

$$C = \sum_{k=1}^{Nc} C_{kk} \tag{8}$$

$$p_k = \sum_{j=1}^{Nc} C_{kj} \tag{9}$$

Recall measurement offers information regarding the capability of the system to correctly predict the phonemes in the dataset and is computed using the following equation:

$$Recall = \frac{C}{C + \sum_k^{Nc} fn_k} \tag{10}$$

where $fn_k$ is the number of times the class $k$ was predicted as one of the other classes:

$$fn_k = \sum_{i=1,\ i \neq k}^{Nc} C_{ki} \qquad (11)$$

## 4. Results

This paper aimed to develop an intelligent subject's shared ISR system for differentiating seven phonemes and four words of the Kara One database. The signals from the database were preprocessed, in order to obtain a better quality of the data and were further introduced to a feature extraction stage, with the purpose of extracting the main information hidden in the EEG signals regarding imaginary speech. The features used were based on the inter-channel covariation in frequency domain, a method of feature extraction first introduced in the ISR domain in the study [17], which has the main advantage of encoding the variability of the electrodes when a stimulus is sent by the brain to the motor neurons involved in the process. Another advantage of the method consists of eliminating the possible delays of brain impulse over the channels, due to the computation in frequency domain, with respect to the time domain, first introduced to imaginary speech by Pramit Saha and Sidney Fels in their study [12]. In the classification stage, we used a 2D CNNLSTM neural network in order to connect both the spatial and temporal correlations between the different windows and the electrodes.

During the development of the system, we also studied the system performances for different regions of the brain, frontal (F), central (C), occipital (O), and different areas, left (L) and right (R) and combinations between them:

- FC = frontal and central;
- CO = central and occipital;
- FCO = frontal, central and occipital;
- Frontal left and right;
- Central left and right;
- Occipital left and right.

To these regions, we added an ASBA (anatomical brain areas involved in speech) region. Considering the nowadays inclination to develop systems that can be easily transposed to a portable device, reducing the number of channels can be a big breakthrough towards this aspiration. By using only the electrodes that are directly responsible for the speech production, the device is more likely to be accepted by the users. Another advantage of selecting the channels consists of reducing the memory and the execution time of the system.

### 4.1. CNNLSTM vs CNN

This study focused on highlighting the advantages of the 2D CNNLSTM neural network in recognizing imaginary phonemes and words using the inter-channel cross-covariance method in the feature extraction stage. The results achieved were compared with the ones recorded in study [17], where the same feature extraction method was used with a CNN neural network as classifier. The results comparison can be observed in Table 3.

**Table 3.** The CNNLSTM neural network performances presented in comparison to CNN neural network, described in [17].

|  | Accuracy | Balanced Accuracy | Kappa | Recall |
|---|---|---|---|---|
| CNN [17] | $0.3758 \pm 0.004$ | $0.3749 \pm 0.004$ | $0.3132 \pm 0.004$ | $0.3750 \pm 0.004$ |
| CNNLSTM [This study] | $0.4398 \pm 0.004$ | $0.4392 \pm 0.004$ | $0.3837 \pm 0.004$ | $0.4398 \pm 0.004$ |

The CNNLSTM neural network managed to increase the accuracy from 0.3758 to 0.4398, according to Table 3, when the same preprocessing and feature extraction processing chain was followed. This difference in accuracy is due to the ability of the LSTM neural network to learn the long-term dependencies of time series in combination with the 2D CNN neural network, which encodes the spatial correlation between the electrodes too.

The mean confusion matrix of the k-folds is presented in Figure 5 and reveals that there is hardly any confusion between the phonemes and words, as expected after the visualization of the LDA dimension reduction results in Figure 2.

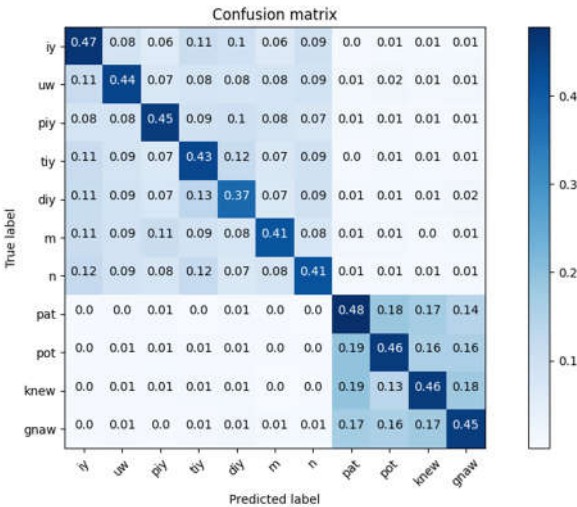

**Figure 5.** The confusion matrix obtained for all eleven classes of KODB when using CNNSLTM as neural network.

An analysis of the distances between the mean of each class for every phoneme and word was conducted in Tables 4 and 5, with the cross-covariance computed in frequency domain features (Table 4) and for the features computed by the CNNLSTM neural network after the first CNNLSTM layer (Table 5). This analysis aims to highlight the role of the CNNLSTM layers in the feature extraction adaptive process, designed to transform the input space and to increase the separability between the classes.

**Table 4.** The computed Euclidean distances between the mean of each class and every phoneme and word of the KODB for the cross-covariance features in frequency domain (the features were normalized before the computation of the Euclidean distance).

|        | /iy/ | /uw/ | /piy/ | /tiy/ | /diy/ | /m/  | /n/  | pat  | pot  | knew | gnaw |
|--------|------|------|-------|-------|-------|------|------|------|------|------|------|
| /iy/   | 0    | 0.96 | 1.20  | 0.97  | 1.02  | 1.46 | 1.21 | 3.40 | 3.45 | 3.19 | 3.00 |
| /uw/   | 0.96 | 0    | 1.44  | 1.16  | 1.16  | 1.61 | 1.18 | 3.34 | 3.49 | 3.17 | 2.97 |
| /piy/  | 1.20 | 1.44 | 0     | 0.97  | 1.26  | 1.41 | 1.15 | 3.44 | 3.38 | 3.23 | 3.05 |
| /tiy/  | 0.97 | 1.16 | 0.97  | 0     | 0.94  | 1.26 | 1.03 | 3.23 | 3.27 | 3.04 | 2.84 |
| /diy/  | 1.02 | 1.16 | 1.26  | 0.94  | 0     | 1.18 | 1.21 | 3.08 | 3.17 | 2.88 | 2.70 |
| /m/    | 1.46 | 1.61 | 1.41  | 1.26  | 1.18  | 0    | 1.12 | 2.81 | 2.80 | 2.67 | 2.43 |
| /n/    | 1.21 | 1.18 | 1.15  | 1.03  | 1.21  | 1.12 | 0    | 2.95 | 2.96 | 2.83 | 2.57 |
| pat    | 3.40 | 3.34 | 3.44  | 3.23  | 3.08  | 2.81 | 2.95 | 0    | 1.10 | 0.65 | 0.93 |
| pot    | 3.45 | 3.49 | 3.38  | 3.27  | 3.17  | 2.80 | 2.96 | 1.10 | 0    | 1.07 | 0.98 |
| knew   | 3.19 | 3.17 | 3.23  | 3.04  | 2.88  | 2.67 | 2.83 | 0.65 | 1.07 | 0    | 0.93 |
| gnaw   | 3.00 | 2.97 | 3.05  | 2.84  | 2.70  | 2.43 | 2.57 | 0.93 | 0.98 | 0.93 | 0    |

**Table 5.** The computed Euclidean distances between the mean of each class and every phoneme and word of the KODB for the features computed by the CNNLSTM first layer (the features were normalized before the computation of the Euclidean distance).

|  | /iy/ | /uw/ | /piy/ | /tiy/ | /diy/ | /m/ | /n/ | pat | pot | knew | gnaw |
|---|---|---|---|---|---|---|---|---|---|---|---|
| /iy/ | 0 | 8.74 | 9.91 | 8.03 | 8.11 | 10.81 | 9.67 | 39.12 | 33.64 | 35.06 | 28.91 |
| /uw/ | 8.74 | 0 | 11.07 | 9.89 | 8.54 | 12.10 | 9.33 | 36.45 | 31.48 | 32.62 | 26.71 |
| /piy/ | 9.91 | 11.07 | 0 | 8.07 | 9.09 | 10.66 | 9.28 | 38.71 | 32.97 | 34.73 | 28.92 |
| /tiy/ | 8.03 | 9.89 | 8.07 | 0 | 7.67 | 8.97 | 8.33 | 38.54 | 32.96 | 34.51 | 28.35 |
| /diy/ | 8.11 | 8.54 | 9.09 | 7.67 | 0 | 8.90 | 9.13 | 37.10 | 31.53 | 33.07 | 26.95 |
| /m/ | 10.81 | 12.10 | 10.66 | 8.97 | 8.90 | 0 | 9.20 | 37.10 | 30.99 | 33.28 | 26.76 |
| /n/ | 9.67 | 9.33 | 9.28 | 8.33 | 9.13 | 9.20 | 0 | 37.37 | 31.42 | 33.46 | 26.93 |
| Pat | 39.12 | 36.45 | 38.71 | 38.54 | 37.10 | 37.10 | 37.37 | 0 | 11.24 | 7.14 | 13.92 |
| Pot | 33.64 | 31.48 | 32.97 | 32.96 | 31.53 | 30.99 | 31.42 | 11.24 | 0 | 8.79 | 8.88 |
| knew | 35.06 | 32.62 | 34.73 | 34.51 | 33.07 | 33.28 | 33.46 | 7.14 | 8.79 | 0 | 10.36 |
| gnaw | 28.91 | 26.71 | 28.92 | 28.35 | 26.95 | 26.76 | 26.93 | 13.92 | 8.88 | 10.36 | 0 |

To compute the Euclidean distance between the mean of each class, first the input matrix was reshaped into a matrix with the size N × N$_{features}$, where N is the number of input vectors, and N$_{features}$ is the total number of features corresponding to N$_{window}$ × N$_{channels}$ × N$_{depth}$ for the cross-covariance matrix in frequency domain and N$_{window}$ × N$_{channels}$ × N$_{channels}$ × N$_{depth}$ for the features computed by the first layer of the CNNLSTM neural network. All features were normalized in the range [0, 1], using Equation (12), before computing the Euclidean distances (13).

$$X_{ij}^{norm} = \frac{X_{ij} - \min_i X_{ij}}{\max_i X - \min_i X_{ij}} \tag{12}$$

where $X_{ij}^{norm}$ is the input vector $X_{ij}$, $i = \overline{0, \ N-1}$, $j = \overline{0, \ N_{features}-1}$, $\min_i X_{ij}$ is the minim value of the vector $i$ for the feature $j$, and $\max_i X_{ij}$ is the maximum value of the vector $i$ for the feature $j$.

The Euclidean distance between the mean of utterance $u1$, and utterance $u2$ was computed as:

$$D = \sqrt{\sum_{j=0}^{N_{features}-1} \left(\mu_{u1,j} - \mu_{u2,j}\right)^2} \tag{13}$$

where $\mu_{u,j}$ is equivalent to the mean of all utterance u, computed as:

$$\mu_u = \frac{1}{N_u} \sum_{i=0, \ X_i \in u}^{N_u - 1} X_i \tag{14}$$

It can be seen from Tables 4 and 5 that the Euclidean distances of the means increased significantly after the first convolutional layer. This growth highlights the important role of the convolutional layer in transforming the feature space into a space with a better separability between classes.

Tables 6 and 7 summarizes the minimum and maximum of the computed Euclidean distance for the phonemes and words, together with the minimum and maximum of the standard deviation of the features to emphasize that, even if the averages move away from

each other, the standard deviation has not undergone significant changes. The standard deviation of each feature for each class was computed using the following equation:

$$\sigma_u = \sqrt{\frac{1}{N_u} \sum_{i=1,\ X_i \in u}^{N_u-1} (X_i - \mu_u)^2} \tag{15}$$

where $\sigma_u$ is the standard deviation for each feature for the utterance $u$.

**Table 6.** The minimum and maximum values of the Euclidian distances and the minimum and maximum of standard deviation over the features for the cross-covariance in frequency domain.

| Euclidean Distance | | Measured between Classes | Std. Dev. of Feature Components Limits | |
|---|---|---|---|---|
| | | | $\sigma_{min}$ | $\sigma_{max}$ |
| Min. | 0.6543 | pat | 0.0754 | 0.2448 |
| | | knew | 0.064 | 0.2367 |
| Max. | 3.4864 | /uw/ | 0.0491 | 0.2166 |
| | | pot | 0.0671 | 0.236 |

**Table 7.** The minimum and maximum values of the Euclidian distances and the minimum and maximum of standard deviation over the features computed after the first CNNLSTM layer.

| Euclidean Distance | | Measured between Classes | Std. Dev. of Feature Components Limits | |
|---|---|---|---|---|
| | | | $\sigma_{min}$ | $\sigma_{max}$ |
| Min. | 7.1444 | pat | 0.0145 | 0.3034 |
| | | knew | 0.0146 | 0.3075 |
| Max. | 39.1189 | /uw/ | 0.0126 | 0.2769 |
| | | pot | 0.0145 | 0.3034 |

### 4.2. Brain Areas Analysis

The second study conducted in this paper was based on evaluating the performance of the system when using a smaller number of electrodes from different regions of the brain. Table 8 details the region selected for the study, along with the electrodes used for that region.

**Table 8.** The analyzed brain areas for channels reduction with the corresponding electrodes from the 10-20 system for electrode positioning.

| | Frontal (F) | Central (C) | Occipital (O) | Anatomical Speech Brain Areas (ASBA) |
|---|---|---|---|---|
| Left (L) | FP1, FPZ, AF3, F7, F5, F3, F1, FZ, FCZ, FC1, FC3, FC5, FT7 | FT7, FC5, FC3, FC1, FCZ, CZ, C1, C3, C5, T7, TP7, CP5, CP3, CP1, CPZ | P7, P5, P3, P1, PZ, POZ, PO3, PO5, PO7, O1, OZ | AF3, F7, F5, F3, F1, FZ, FCZ, FC1, FC3, FC5, FT7, T7, C5, C3, C1, CZ |
| Right (R) | FP2, FPZ, AF4, FZ, F2, F4, F6, F8, FT8, FC6, FC4, FC2, FCZ | FCZ, FC2, FC4, FC6, FC8, T8, C6, C4, C2, CZ, CPZ, CP2, CP4, CP6, TP8 | PZ, P2, P4, P6, P8, PO8, PO6, PO4, POZ, OZ, O2 | AF4, FZ, F2, F4, F6, F8, FT8, FC6, FC4, FC2, FCZ, CZ, C2, C4, C6, T8 |

The regions were selected at first based on the major brain areas corresponding to the EEG electrodes, i.e., frontal, central, and occipital areas, and afterwards based on

the anatomical brain areas involved in speech conceptualization and articulatory plans, initiation, and coordination of the motor stimulus to be sent to the effectors. Regarding the anatomical brain areas involved in speech, is well-known that the Broca area has an important role in speech production [23]: the primary motor cortex generates the signals that control the execution of the movements and was discovered to also relate with the motor imagination [24], and the secondary motor area is responsible for motor planning [25]. The next step was to identify the spatial position of each anatomical structure identified to be involved in speech, regarding to the electrodes positioned in the 10-20 system. The Broca area is positioned in the posterior half of the inferior frontal gyrus; the primary motor cortex corresponds to the precentral gyrus, and the secondary motor cortex is mostly identified as Brodmann area 6, which is positioned in the precentral gyrus, the caudal superior frontal gyrus, and the caudal middle central gyrus. These anatomical spatial structures were identified as electrodes in the 10-20 positioning system by the researcher L. Koessler et al. in [26] as the channels selected for the anatomical speech brain areas detailed in Table 8.

For a better perspective of the analyzed brain areas, Figure 6 presents the 10-20 system for the electrode positioning used to acquire the signals from Kara One database having the selected regions colored differently.

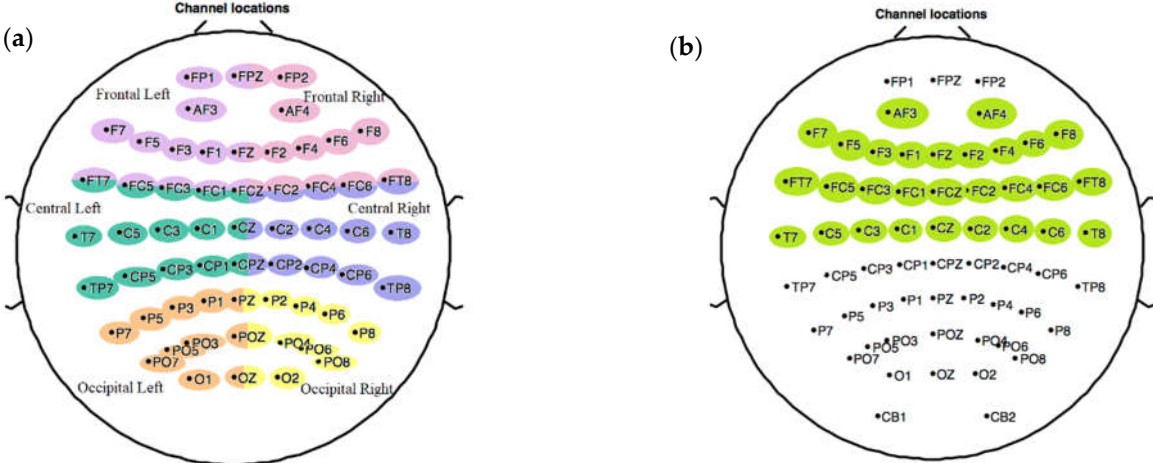

**Figure 6.** The representation of the different brain areas selected for the channels dimension reduction corresponding to the (**a**) major areas of the EEG represented with different colors and (**b**) ASBA.

The obtained results for each brain area and the combinations of the brain areas studied in this paper are presented in Table 9.

**Table 9.** The mean of k-folds (k = 4) accuracy of the classification results obtained using different areas and combinations of different areas as CNNLSTM input; F—frontal, C—central, O—occipital, FC—frontal and central, CO—central and occipital, FCO—frontal, central, and occipital, ASBA—anatomical speech brain areas, L—left, and R—right.

|   | F | C | O | FC | CO | FCO | ASBA | F | C | O | ASBA |
|---|---|---|---|---|---|---|---|---|---|---|---|
| L | 0.2846 ± 0.005 | 0.2892 ± 0.005 | 0.2297 ± 0.002 | 0.3372 ± 0.002 | 0.3132 ± 0.002 | 0.3518 ± 0.001 | 0.3096 ± 0.006 | 0.3582 ± 0.006 | 0.3678 ± 0.005 | 0.2606 ± 0.006 | 0.4027 ± 0.002 |
| R | 0.3028 ± 0.003 | 0.3032 ± 0.003 | 0.2272 ± 0.004 | 0.3386 ± 0.007 | 0.3189 ± 0.001 | 0.3566 ± 0.003 | 0.3305 ± 0.001 | | | | |

As expected, after selecting fewer channels corresponding to different brain areas of interest, the system classification accuracy decreased. The best accuracy was achieved when using the electrodes from the identified anatomical speech brain areas, according

to the specialized literature, from both the left and right hemispheres, reaching a value of approx. 40%. Although there was a decrease in performance, compared to using all the available channels, and there were advantages regarding the computation, memory, costs, and even the comfort of using a smaller number of electrodes.

### 4.3. Complexity and Memory Analysis

The tendency of an ISR system is to perform with the highest possible accuracy on a portable device with limited resources. There will always be a tension between the performance and the complexity when developing an ISR system. In this section of the paper, we focused on a study regarding the complexity and memory of the developed system.

When it comes to the complexity and memory of an intelligent system, the primary resource consumer is generally the neural network. In the case of a CNN neural network with long-term memory, the complexity is given by the long-term memory convolutional layer, due to the need of the implied gates output computation, i.e., the input gate, the forgetting gate, and the output gate, considering the 2D spatial correlation in this process, too. For the 3D convolution corresponding to the input tensor ($N_{lines} \times N_{columns} \times N_{channels}$) and the kernel ($k_{lines} \times k_{columns} \times k_{channels}$), the complexity can be measured as:

$$O((N_{lines} + k_{lines})(N_{columns} + k_{columns})(N_{channels} + k_{channels}) \log((N_{lines} + k_{lines})(N_{columns} + k_{columns})(N_{channels} + k_{channels}))) \tag{16}$$

when using the fast Fourier transform, as described in [27].

In our case, the $N_{lines} = N_{columns}$, $k_{lines} = k_{columns}$, and $N_{channels} = k_{channels}$, so the Equation (16) becomes:

$$O\left(2(N + k)^2 N_{ch} \log\left(\left(2(N + k)^2 N_{ch}\right)\right)\right) \tag{17}$$

where $N$ is the input lines/columns, $k$ is the kernel dimension, and $N_{ch}$ is the number of input channels.

However, for a ConvLSTM2D layer, this computation (described in Equation (17)), will be made for all the filters of the layer ($N_f$), for all windows ($N_w$), and for all gates, plus the computation of the last cell output (a total of four computations). Finally, the complexity can be approximated to:

$$O\left(4N_w\left(2N^2 N_{ch}\right) \log\left(\left(2N^2 N_{ch}\right)\right) N_f\right) \tag{18}$$

The details regarding the complexity for all system stages and for all specific layers of the used neural network are presented in Table 10, along with the memory consumption and the execution time measured using an AMD Ryzen 7 4800HS CPU.

**Table 10.** The complexity and memory of the system detailed for each step taken for system development.

| System Stages | | Complexity | Memory | Time [s] |
|---|---|---|---|---|
| Feature Extraction | FFT | $O(Nf \times N \times M\log M)$ | ~23 MB | <0.001 |
| | COV | $O(Nf \times N^2)$ | ~24 MB | <0.001 |
| CNNLSTM | ConvLSTM2D-64 | $O(4 \times 4 \times (2N^2) \times \log(2N^2) \times 64)$ | ~2.12 GB | 0.0819 |
| | ConvLSTM2D-128 | $O(4 \times 4 \times (2N^2 \times 64) \times \log(2N^2 \times 64) \times 128)$ | | |
| | Dense-64 | $O(4 \times N^2 \times 128 \times 64)$ | | |
| | Dense-11 | $O(64 \times 11)$ | | |

## 5. Discussion

This paper aimed to develop an intelligent system for the imaginary speech recognition of seven phonemes and four words from the Kara One database. To achieve our goal, we passed the signals from the database through a preprocessing stage, in which an expert analyzed the imaginary speech epochs and eliminated the ones with high noises. Afterwards, in the feature extraction stage, we computed the cross-covariance between the channels in the frequency domain to conserve the channels connections in a compact matrix. In this stage, we showed, by using the LDA algorithm for feature visualization in a 2D feature space, that this method is better for decoding the features than using the signals in time or the cross-covariance applied over the channels in time. We used, as a classifier, an CNNLSTM neural network and concluded that it performed better than a CNN (in comparison to paper [17]), due to the consideration of not only the space correlations, but of the time variations, too. We were looking to develop a system that can easily be passed to a low-cost portable device, so we also studied the possibility of using a smaller number of electrodes divided into different brain regions. Finally, we also studied the complexity of the algorithm and the time required for the system to decide the membership of an input.

### 5.1. LDA for Feature Extraction Analysis

In the feature extraction stage, the signals were passed through a feature reduction algorithm used for the visual analysis of the three different features: the signal in time without processing, the cross-covariance in time, and the cross-covariance in the frequency domain. We observed that, when using the cross-covariance in the frequency domain, the features were clustered into the desired classes, however overlapped, but obtaining a good distinction between the phonemes and the words. We also observed that the distances between the means got higher using this type of feature extraction. When using the cross-covariance in time domain, the phonemes and the words were distributed in clusters, but almost completely overlapped, which made the classification harder. The computed means of each class was higher, compared to the raw feature signal, but the separability between the phonemes and words was lost. We observed that the means of the raw signals were close to each other for phonemes and for words, but a distinction between the two was clearly made.

### 5.2. CNNLSTM vs. CNN

The main advantage of a LSTM neural network is its capability of memorizing long-term dependencies. This neural network ability comes in handy when non-stationary time variant signals, such as EEG, are analyzed. In addition to the LSTM long-term memory, the spatial dependencies of the EEG signals were taken into consideration by adding CNN to the LSTM layer. This junction helped the neural network to learn both spatial and time-variant features, increasing the accuracy of the system from 0.37, obtained with a CNN architecture with similar parameters up to 0.43.

The mean confusion matrix of the k-folds presented in Figure 5 reveals that there is hardly any confusion between the phonemes and words, as expected after the visualization of the LDA dimension reduction results in Figure 2. Confusions were made between the phonemes /tiy/ and /diy/, with a relatively higher percent, compared to the rest of the phonemes. This behavior can be explained by the similar mechanisms of the pronunciation of the sound "t" and the sound "d". For both utterances, the vocal tract maintains almost the same position, and the sound is being produced by presence or absence of vocal cord vibration. Major confusions of approx. 19% were also made between the words "pat" and "pot", for the same reason of similar mechanism of pronunciation.

In Table 4, we computed the Euclidean distance between the mean of each phoneme and word analyzed for the cross-covariance in the frequency domain. We observed that these distances are smaller than the ones computed for the extracted features obtained after the first CNNLSTM layer (Table 5), meaning that, after the signal was processed by the first

neural network layer, the feature space of the utterances was modified, and the features became more separable.

### 5.3. Brain Area Analysis

The final goal of an automatic ISR system is to obtain the best possible accuracy using a portable device with limited resources. Considering this, we further studied the system behavior by using a smaller number of electrodes, located in specific areas of the brain, for classifications. This approach enhanced the portability of the device and decreased the needed resources used for development, but with the cost of accuracy, as can be seen in Table 9.

When the number of channels was reduced, the accuracy dropped, as well. However, when using the electrodes positioned on the anatomical regions of the brain responsible for speech intention and production, the accuracy of the system reached 0.40, a value with a drop of only 3%, in comparison to all channel accuracy. This means that 93% of speech information is concentrated in these channels, and only 7% of information is distributed to the parietal and occipital regions. The main advantage of using only the channels corresponding to the anatomical structures of the brain regarding speech is that the number of electrodes is reduced, in this case by more than half, from 62 to 29, which is a big computational and cost gain.

Important information obtained after the study of the brain areas involved in imaginary speech recognition is that the data acquisition using the visual stimulus appearing on a prompt does not affect the study because the occipital area is less involved in the decision-making of the system. This is due to the two-second period introduced in the protocol design, the time between the prompt display and the actual imagination of the phoneme.

### 5.4. Complexity and Memory Analysis

An important aspect when developing an ISR system is considering the complexity of the algorithm and the memory used. Usually, the main consummator of the resources is the neural network, as can also be seen in Table 10. The maximum number of operations is given by the second layer of the CNNLSTM neural network and is to the order of approx. $O$ ($6.3 \times 10^9$). However, the execution time for a decision is under 100 ms, even when using all the channels in the computation, which means that it can still be implemented in a real-time device.

When reducing the number of electrodes, we can see in Table 11 an important decrease in the time execution, as well. An alternative to an all-electrode system can be the ASBA electrodes system, in which the time of execution considerably dropped from 80 ms to 20 ms, with a drop in accuracy of only 3%. This alternative is better from other points of view, too, such as increasing the user comfort when using the device and decreasing the cost of the final product.

**Table 11.** The mean execution time needed to obtain a response from the system for all groups of channels analyzed in the paper; L—left, R—right, ASBA—anatomical speech brain areas.

|  | No. Channels | Time [s] | Accuracy [%] |
| --- | --- | --- | --- |
| All channels | 62 | 0.0826 | 43.98 |
| L/R Frontal | 13 | 0.0046 | 28.46/30.28 |
| L/R Central | 15 | 0.0057 | 28.92/30.32 |
| L/R Occipital | 11 | 0.0035 | 22.97/22.72 |
| L/R Frontal + Central | 23 | 0.0124 | 33.72/33.86 |
| L/R Central + Occipital | 26 | 0.0155 | 31.32/31.89 |

**Table 11.** *Cont.*

|  | No. Channels | Time [s] | Accuracy [%] |
|---|---|---|---|
| L/R Frontal + Central + Occipital | 34 | 0.0258 | 35.18/35.66 |
| L/R ASBA | 16 | 0.0068 | 40.27 |
| L + R Frontal | 23 | 0.0167 | 35.82 |
| L + R Central | 27 | 0.0156 | 36.78 |
| L + R Occipital | 16 | 0.0089 | 26.06 |
| L + R ASBA | 29 | 0.0209 | 40.27 |

When it comes to the memory usage, the system has its limitation because it needs at least 2 GB of memory only to retain the weights, due to the gates and the long-term memory implication of the LSTM network. However, the LSTM addition to the system improved the overall classification and added great value to the final system.

## 6. Conclusions

This paper aimed to develop a subject's shared system for recognizing seven phonemes and four words collected during imaginary speech from the Kara One Database. To achieve the proposed goal, the database was preprocessed and features were computed, in order to decode the hidden information regarding imaginary speech. The features were based on computing the cross-covariance in the frequency domain and were introduced to a CNNLSTM neural network for final classification in the desired classes.

During our research, we observed that, by computing the cross-covariance in frequency domain, the feature space turned into a space where the utterances could be separated easily. This conclusion was drawn after analyzing the feature space in two dimensions by computing the LDA algorithm for feature reduction.

Another feature comparison was made between the extracted features and the features obtained after the first CNNLSTM layer. Tables 4 and 5 represent the Euclidean distances between the mean of each class, and it can be easily seen that these distances increased after the CNNLSTM layer. This suggests that the feature space of the input vectors changed, so that the classes became more separable. This is one of the advantages of deep neural networks, especially CNN networks, which use the first layers for feature extraction before the actual classification of the data.

This paper also showed an improvement for the system performance when using a CNNLSTM neural network, with respect to CNN. The accuracy increased from 37% to 43% when using CNNLSTM and the same processing chain of the database. The advantage of CNNLSTM is the consideration of both the spatial and temporal connections. CNN uses the convolution for the spatial connection between the channels, while LSTM brings the long-term memory that is vital for the non-stationary time variant signals, such as EEG.

The proposed system considers the portability of a possible real-time portable device. Therefore, we also studied the system's behavior when reducing the number of channels for classification. We separated the electrodes into main areas, i.e., frontal, central, and occipital, studied the areas involved in imaginary speech production, and selected the channels accordingly for the anatomical speech brain areas study. We concluded that 93% of the information is concentrated in the anatomical speech brain areas, obtaining an accuracy of 40% for 29 channels used, in comparison to the 62 used in the beginning. Even if the accuracy dropped by 3% using only 29 electrodes, the system brings more advantages in a matter of time for execution, portability, comfort, and cost.

**Author Contributions:** Conceptualization, A.-L.R. and O.G.; Methodology, A.-L.R. and O.G.; Software, A.-L.R.; Data curation, A.-L.R.; Writing – original draft, A.-L.R.; Writing – review & editing, O.G.; Supervision, O.G. All authors have read and agreed to the published version of the manuscript.

**Funding:** This research received no external funding.

**Institutional Review Board Statement:** Ethical approval was obtained from both the University of Toronto and the University Health Network, of which Toronto Rehab is a member.

**Informed Consent Statement:** Consent was obtained from all subjects involved in the study.

**Data Availability Statement:** http://www.cs.toronto.edu/~complingweb/data/karaOne/karaOne.html (accessed on 29 September 2022). https://github.com/LuizaRusnac/BSThinkingOutLoud.git (accessed on 29 September 2022).

**Conflicts of Interest:** The authors declare no conflict of interest.

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
