# Peer review of "Imaginary Speech Recognition Using a Convolutional Network with Long-Short Memory"

_applsci, doi:10.3390/app122211873_

Round 1

Reviewer 1 Report

This paper  has been aimed to develop a subject’s shared system for recognizing seven pho- nemes and four words collected during imaginary speech from Kara One Database. To  achieve the proposed goal, the database has been preprocessed, and features were computed  in order to decode the hidden information regarding imaginary speech. The features have been based on computing the cross-covariance in frequency-domain and were introduced to a CNNLSTM neural network for final classification in the desired classes.

The Authors have been used CNNLSTM neuratl network model to modelling of the  problem.  They  have   got  the improvement of the accuracy that  is incresed  37% to %43.

The proposed system has been considered the portability of a possible real-time portable device. Therefore, they  have  also been studied the system behavior when reducing the number of channels  for classification. They  have separated the electrodes into main areas: Frontal, Central, Occipital, and studied the areas involved in imaginary speech production and selected the channels  accordingly for the anatomical speech brain areas study. They have  concluded that 93% of information was concentrated in the anatomical speech brain areas, obtaining an accuracy of 40%  for 29 channels used, in comparison to 62 used in the beginning. Even if the accuracy  dropped with 3% using only 29 electrodes, the system brings more advantages in matter of time of execution, portability, comfort, and cost.

There is some gaps in the  syudy;

-Have The Authors  used any software? If they have used a softwarer;   what is the name of the softwaere?

-If they used  a software;   what is the CNNLSTM neural network code? İs it  open source or  have the authors  created it? How?

-The Authors have been  explained the problem  as theoretically, but  CNNLSTM neural network model hasnt been explanied the study.  How they have  created the model?  Which steps  have been used to  developing the model?  Have they   tried different neural network model? And how much the  performance of the  tried other models? İf they are  tired  different models and algorithms.

- How the Authors  obtained the  permormance  43%.  What is the scala of the performance?  The Authors must be explaned the  this  performance  ratio.

- When i  critised the  study,  method section is not  enough  fort he readers and  scientist.

-Similartiy repart has 12%  except  bibliograpy.  It can be reduced max %10.

Because of  above comments, mty decision is  Major revision.

Author Response

We appreciate the time dedicated to our paper and we consider that you provided valuable comments. The authors have carefully considered the comments and tried our best to address every one of them.

  1. For developing the system, we used Python as programming language together with the open-source libraries: NumPy, TensorFlow, scikit learn.
  2. The development of the CNNLSTM layers can be found in TensorFlow and is open source
  3. We provided the references for the development of the CNNLSTM neural network both theoretically [1] and for the code source [2]. We also tried CNN neural network with multiple architectures and hyperparameters in our previous paper published in Sensors [3]. Based on the best obtained results, we chose the given CNNLSTM architecture and focused in this paper on selecting the electrodes for a system with a lower complexity.
  4. We added in the paper the explanation of all the performance metrics used to evaluate the system
  5. We hope that we managed to improve the method section. Otherwise, we are open to any suggestions that might improve this section.
  6. Regarding the similarity report, we did our best to drop the 12% to 10%, but the major similarities are with our last paper [3] and words like “cross-covariance in frequency domain”, “cross-covariance in time domain”, the phonemes and words used for the imaginary speech database and the CNN layers and architecture which are repeated over the paper are needed in both studies.

[1] X. Shi, Z. Chen, H. Wang, D.-Y. Yeung, W. Wong, and W. Woo, “Convolutional LSTM Network: A Machine Learning Approach for Precipitation Nowcasting,” 2015, doi: 10.48550/ARXIV.1506.04214

[2] M. Abadi et al., “Tensorflow: A system for large-scale machine learning,” in Symposium on Operating Systems Design and Implementation, 2016, pp. 265–283

[3] A.-L. Rusnac and O. Grigore, “CNN Architectures and Feature Extraction Methods for EEG Imaginary Speech Recognition,” Sensors, vol. 22, no. 13, p. 4679, Jun. 2022, doi: 10.3390/s22134679

Reviewer 2 Report

The introduction and state-of-the-art are well-written. There is a need to define the meaning of the parameters of table 4.1 such as Accuracy, Balanced Accuracy, Kappa, and Recall, and what they describe. Need to clarify the meaning of various phonemes such as /tiy /, /diy/, /pot etc. .The presentation of the results and conclusion are appropriate.  

Author Response

First of all, thank you very much for the time allocated to review the paper.

We added in the paper the meaning of the Table 4.1 parameters. We hope that now the computation of the system performance is clearer.

We also specified the different mechanisms of pronunciation for the chosen phonemes from the database (Lines 187 – 192) to justify the selection of the researchers from Toronto University.

We hope that we managed to address all the observations with clarity. If not, we are open to receive new constructive comments.

Round 2

Reviewer 1 Report

There are some improvment but it is not   enough to  publish  a SCI journal.  My  propposes have not been  completed.  Material and Metods are not enouht to  define and  demonstrated the problem.   

Author Response

First of all, we are very grateful for your contribution in desire of developing the quality of the paper. We consider that each comment conducts to a better and improved version of our final work.

However, we are truly sorry, but we don’t really understand what part of Material and Methods are not proper described. We will be very grateful to you if you could provide us more information, point by point, about what should we add to this section. We would really appreciate this help from you.

For the Classification stage: Is the need of describing better theoretically or with specific equations the CNNLSTM neural network? Is the need of providing the python code used for developing the system? Or maybe the need of a pseudocode relatively to the CNNLSTM neural network?

For the Results section: Is anything we could add to provide a clearer presentation of them? Perhaps a new metric?  Another analysis method?

For the Conclusion section: We would really need some clarifying information here because we don’t really know why the conclusions are note supporting the results.

Again, we are sorry for not understanding what reviewer observations were and need extra help, with point-by-point detailed information about the needed modifications to achieve the desired performance required for the paper. Thank you in advance for the extra time allocated to this task.

Round 3

Reviewer 1 Report

Authors have been re write and  re arranged the manuscript.  Methodolog and  presentation is well enough. So It can be  published  the Journal.